# Morphological and Physiological Changes of *Broussonetia papyrifera* Seedlings in Cadmium Contaminated Soil

**DOI:** 10.3390/plants9121698

**Published:** 2020-12-03

**Authors:** Wan Zhang, Yunlin Zhao, Zhenggang Xu, Huimin Huang, Jiakang Zhou, Guiyan Yang

**Affiliations:** 1Hunan Research Center of Engineering Technology for Utilization of Environmental and Resources Plant, Central South University of Forestry and Technology, Changsha 410004, China; zw@csuft.edu.cn (W.Z.); zyl8291290@163.com (Y.Z.); Huanghm1991@163.com (H.H.); zjk@csuft.edu.cn (J.Z.); 2College of Forestry, Northwest A & F University, Yangling 712100, China; yangguiyan@nwsuaf.edu.cn

**Keywords:** *Broussonetia papyrifera*, cadmium contaminated soil, antioxidant enzyme, anatomic structure

## Abstract

*Broussonetia papyrifera* is a widely distributed economic tree species, and it is also a pioneer species in adverse environments. In order to investigate the growth and adaptation mechanism of *B. papyrifera* under cadmium (Cd) contaminated soil, potted experiments were used with six-month treatments to study Cd enrichment and the transportation, morphological and physiological characteristics of *B. papyrifera* tissues. The results showed that Cd mainly accumulated in the root when the Cd concentration was high (14.71 mg/kg), and the root biomass was significantly reduced by Cd stress although Cd promoted the growth of seedlings. The bioconcentration factors (BCF) increased with the increase in Cd concentration, and reached the maximum value of 0.21 at 14.71 mg/kg. On the contrary, translocation factor (TF) decreased significantly at 8.28–14.71 mg/kg Cd concentration. Cd not only led to the loose arrangement of the xylem vessels of leaves, but also changed the chlorophyll content. However, *B. papyrifera* could synthesize organic solutes such as soluble protein, soluble sugar and proline to reduce the intracellular osmotic potential. Our study proved that *B. papyrifera* has good tolerance to Cd stress and is a pioneer tree species for soil and ecological environment restoration.

## 1. Introduction

Due to the impact of industrialization and geochemical activities, heavy metal pollution in soil has become a serious environmental problem that limits crop production and even threatens human health through the food chain [1,2]. Cadmium (Cd) is a natural element in the earth’s crust and is usually combined with other elements (such as oxygen, chlorine or sulfur) to form minerals. Over the past two centuries, a large amount of Cd has been released and accumulated in the soil environment [3,4]. The traditional method of reducing the toxicity by replacing the soil contaminated with heavy metals is highly expensive, and the placement of contaminated soil might become an urgent problem to be solved, which could even cause secondary pollution [5]. At present, phytoremediation has become a more promising method of environmental restoration based on biological extraction and filtration [6,7,8,9]. Hyperaccumulators have good application prospects in the remediation of contaminated soil, such as the first known arsenic (As) hyperaccumulation plant *Pteris vittata* [10,11,12], the manganese (Mn) hyperaccumulation plant *Phytolacca acinosa* [13,14], *Sedum* [15,16] with super enrichment ability to Cd, etc. However, most of the hyperaccumulators are dwarf and the accumulation of heavy metals will also make the plants grow slowly, which brings great difficulties to their practical application. Woody plants have large biomass and a long growth cycle. Besides the developed root system, the area of stems, branches, and leaves form a larger green space. Moreover, most pollutants would not endanger people’s health by food. Thus, woody plants are considered to be a candidate vegetation source for heavy metal pollution remediation.

In order to cope with different environmental conditions, plants would produce dynamic responses and positive adjustment in their biological characteristics, such as morphological structures, physiological mechanisms and even genetic characteristics [17,18,19]. Changes in plants’ morphologies are the most intuitive response to abiotic stress. Plants could adapt to water stress by reducing stomata frequency, stomata length and reducing leaf thickness, as well as increasing the thickness of the palisade parenchyma [20]. Under the lead stress, the *Artemisia Sacrorum var. Messerschmidtiana* in the mineral area showed stronger tolerance than that in the non-mineral area, which was mainly reflected in the thickening of the stem epidermis, cell wall and xylem tube wall [21]. Under the condition of low Mn concentration (1 mmol/L), the biomass, surface area and root volume of *B. papyrifera* were increased [22]. Of course, the physiological response would not be absent in the process of coping with environmental stress. Plants could resist the toxicity of Cd and Mn by promoting the synthesis of soluble protein and increasing the activities of antioxidant enzyme, while high concentrations of Cd and Mn led to the increase in malondialdehyde (MDA), proline and H_2_O_2_ content, causing membrane lipid peroxidation and severe membrane damage [22,23]. Studies have also shown that Cd could inhibit the photochemical reaction of plant photosynthesis and destroy the metabolism of chlorophyll, and is an effective inhibitor of photosynthesis [6,24,25].

*Broussonetia papyrifera*, also known as paper mulberry, is a deciduous tree or shrub. It is widely distributed around the world, providing possibilities for extensive ecological restoration [26,27]. The *B. papyrifera* also has high economic benefits, its bast cellulose content can be as high as 63.76% [28], which can be blended with cotton [29], and the earliest high quality bark cloth was made of *B. papyrifera* bast [30,31]. The leaves are rich in amino acids, proteins, fats and trace elements, which makes them a good source of feed. In addition, the leaves are covered with fine pubescence and can be used to monitor the air pollution in their growing areas [32]. As a medicinal resource, *B. papyrifera*’s flowers are natural antioxidants and can treat impotence and ophthalmic diseases [33,34]. The fruit extract is not only a traditional medicine for treating cardiovascular diseases, but also contains alkaloids as a drug candidate for treating cancer [35]. Stem bark has a high content of phenolic compounds and is a potential resource for medical pharmaceuticals and anti-inflammatory drugs [36,37]. The root system of the *B. papyrifera* is very developed. Under natural conditions, it is mainly propagated by root sprout, and it is an environmental tree species that prevents soil erosion. Because of its fast-growing and adaptable ability, the tree is a pioneer species in desertification areas, mines, rivers, valleys, droughts and other extreme environments [38,39].

Based on the good economic potentiality and environmental effects, *B. papyrifera* is an important candidate plant for phytoremediation [22]. At present, the research on the adversity of *B. papyrifera* mostly focuses on drought and saline–alkali stresses [40,41,42]. There are few studies on the tolerance mechanism under heavy metal pollution. Among them, Zhao et al. [43] investigated the heavy metal enrichment of heavy metal polluted plants and found that *B. papyrifera* had higher enrichment ability for Pb and Zn, especially the leaf part, and the Cd enrichment ability was also higher than in other species. The manner in which *B. papyrifera* responds to heavy metal stress requires a detailed analysis. In order to explore more details about the response of *B. papyrifera* to Cd stress, the seedling growth, leaf anatomical structure and related physiological indexes of *B. papyrifera* were determined under different Cd stress conditions. The research can not only help understand the mechanism of *B. papyrifera* adaptation to poor environments, but also help to establish a technical ecology restoration system based on the tree.

## 2. Results

### 2.1. Height Changes and the Characteristics of Leaf Phenotype

Within 180 d of Cd stress, the height of *B. papyrifera* seedlings in the soil without Cd solution irrigation (non-irrigated soil) did not change significantly (*p* > 0.05) (Figure 1). The initial height of all *B. papyrifera* seedlings averaged about 40 cm. In order to adapt to the seasonal changes, the plant entered the wintering period and stopped growing at 90 d. At this stage, the growth change of *B. papyrifera* under 5.71 mg/kg Cd concentration was the largest, with an increase of 38.34 ± 5.31 cm. At 180 d, the height of seedlings increased significantly at 11.49 mg/kg and 14.71 mg/kg Cd concentrations, indicating that Cd stress can stimulate the growth of *B. papyrifera* seedlings. After 20 d, the leaf phenotype of *B. papyrifera* seedlings changed (Appendix A: The photo shows the second fully expanded leaf of the seedling from top to bottom), especially at 11.49 mg/kg and 14.71 mg/kg Cd concentrations. There were folds on the surface of the leaves. In addition, at a Cd concentration of 8.28 mg/kg, linear spots appeared at the tip of the leaves.

### 2.2. Changes in Biomass and Moisture Content (MC)

Compared with non-irrigated soil, the results showed that application of Cd solution significantly reduced the root biomass, but had no significant effect on the biomass of the stem and leaf (Table 1). The biomass of root in the 5.07 mg/kg, 5.71 mg/kg, 8.28 mg/kg, 11.49 mg/kg and 14.71 mg/kg experimental groups accounted for 38.11%, 24.27%, 25.12%, 31.85% and 29.99% of whole plant biomass, respectively. In addition, the MC in root and stem of the seedlings was not affected significantly (*p* > 0.05); the MC of the roots was 75–80%, and the stems’ MC was 60–70%.

### 2.3. Cd Content in B. papyrifera Tissues

After 180 d, with the increase in Cd concentration, the Cd contents in the root, stem and leaf of *B. papyrifera* all reached the maximum value at 14.71 mg/kg (Table 2). Under the treatment of high concentration of Cd (14.71 mg/kg), the accumulation of Cd in *B. papyrifera* tissue was as follows: root > leaf > stem, while the accumulation ability of different tissues was leaf > stem > root under the treatment of lower Cd concentrations (5.07 mg/kg and 5.71 mg/kg). The greater the Cd concentration, the greater the BCF value, indicating that *B. papyrifera* has a stronger ability to accumulate Cd at a higher concentration. With the increase in Cd concentration, the value of TF gradually decreased, and it decreased significantly at 8.28–14.71 mg/kg (*p* < 0.05).

### 2.4. Anatomic Structure of Leaves

After two months of Cd stress, the anatomical structures of the *B. papyrifera* leaves treated with different Cd concentrations were observed (Figure 2). The leaves of *B. papyrifera* have relatively complete epidermis (including cuticle, upper epidermis and lower epidermis) in all groups. The palisade tissue was more developed, arranged neatly and tightly, and some dense dark spots were visible inside, containing more chloroplasts. Developed palisade tissue can greatly improve the photosynthesis efficiency of *B. papyrifera* and promote its rapid growth. At the same time, the sponge tissue distribution was relatively uneven, the arrangement was loose, and the chloroplast was less affected by Cd. Compared with low Cd concentration (5.07 mg/kg), the tightness of the palisade tissue was loose under the Cd treatments of 5.71 mg/kg, 8.28 mg/kg and 11.49 mg/kg, and the arrangement was disordered, while it was alleviated at 14.71 mg/kg. The xylem vessels in 5.07 mg/kg were arranged closely. With the concentration increasing, the vessels were arranged loosely, which directly affects the absorption of inorganic salt ions and water in *B. papyrifera*. The phloem fiber was tough and has strong flexing resistance. It is mainly responsible for the transport of organic matter (sugar, protein, etc.) and some mineral element ions. The phloem of the *B. papyrifera* leaves after Cd stress was faintly visible, but its thickness decreased with the increase in stress concentration, which was most obvious at 8.28 mg/kg.

### 2.5. Physiological Characteristics of B. Papyrifera Seedlings

#### 2.5.1. Changes in Soluble Protein Content, Soluble Sugar Content and Proline Content

The soluble protein content, soluble sugar content and proline content of *B. papyrifera* leaves have changed over time in all groups (Figure 3). Two-way analyses of variance showed that the soluble protein content of *B. papyrifera* seedlings was hardly affected by the Cd concentration variable, while the stress time could significantly affect the soluble protein content (*p* < 0.001) (Figure 3A). At 12 h, the soluble protein content decreased significantly under 14.71 mg/kg Cd concentration (*p* < 0.05). As the stress time prolonged, at 10 d and 20 d, the soluble protein content increased with the increase in the Cd concentration. On the other hand, at 30 d, the soluble protein content in all treatment groups increased significantly (*p* < 0.05) (Figure 3A), while it first decreased and then increased at the low Cd concentration (5.07 mg/kg).

Soluble sugar, as a carbohydrate, is the main raw material and storage material for plant metabolism. For the low Cd concentration (5.07 mg/kg and 5.71 mg/kg), the soluble sugar content had no significant difference except for 20 d (*p* < 0.05, Figure 3B). With the increase in Cd concentration, the soluble sugar content decreased at 12 h, while at 10 d, 20 d and 30 d, the soluble sugar content increased first and then decreased. At 10 d and 20 d, both reached the maximum under 8.28 mg/kg Cd concentration, but there was no significant difference between the 14.71 mg/kg and 5.07 mg/kg Cd treatment groups. In the same Cd treatment group, the soluble sugar content increased significantly after 20 d (Figure 3B). In the non-irrigated soil, affected by the Cd content in the soil, the soluble sugar content decreased significantly at 10 d (*p* < 0.05), but it was not significantly different after 20 d.

The change of proline content in *B. papyrifera* leaves showed that in a short period of time (12 h), the proline content increased first and then decreased with the increase in Cd concentration in soil. However, there was no significant difference between low Cd concentrations (5.07 mg/kg and 5.71 mg/kg) (*p* > 0.05) (Figure 3C). The data ranges from 12.34 ± 0.31 μg/g FW to 24.25 ± 1.27 μg/g FW. With the extension of time, even if the concentration was very low, it would be significantly different from the non-irrigated soil. At 20 d, the content of proline in the 5.71 mg/kg Cd concentration reached 62.69 ± 3.16 μg/g FW, which was 42.42 ± 2.04 μg/g FW higher than the 5.07 mg/kg Cd concentration (*p* < 0.05). At 30 d, the content of proline in the 5.71 mg/kg Cd concentration was also significantly higher than that in the 5.07 mg/kg Cd concentration (*p* < 0.05). In the different groups, the proline content increased with the extension of time, and the change was most significant at 10 d (Figure 3C).

#### 2.5.2. Changes in MDA Content and Antioxidant Enzyme Activity

In all groups, MDA content, SOD, POD and CAT activities increased with prolonged stress time (Figure 4). In a short period of time (12 h), the content of MDA in *B. papyrifera* leaves was significantly higher in high Cd concentrations (11.49 mg/kg and 14.71 mg/kg) than in low Cd concentrations (5.07 mg/kg and 5.71 mg/kg) (Figure 4A). With the extension of stress time, there was no significant difference between low Cd concentration and high Cd concentration. The results showed that high Cd concentration has a greater impact on *B. papyrifera* seedlings in a short time and more ROS accumulated in *B. papyrifera*. Increased antioxidant enzyme activity could reduce oxidative damage caused by active oxygen accumulation. At 12 h, the activities of SOD, POD and CAT increased with the increasing of the Cd concentration, and the difference was significant from 5.07 mg/kg Cd concentration (*p* < 0.05) (Figure 4B–D). Among them, at 14.71 mg/kg Cd concentration, the activities of SOD, POD and CAT were 729.71 ± 4.13 U/mg prot, 329.40 ± 5.49 U/mg prot and 110.33 ± 6.24 U/mg prot, respectively, which were more than twice as much as those at 5.07 mg/kg Cd concentration with 298.24 ± 7.47 U/mg prot, 153.77 ± 3.10 U/mg prot and 36.75 ± 1.64 U/mg prot, respectively. With the extension of the stress time, the activities of the three antioxidant enzymes in the high Cd concentration decreased. At 10 d and 20 d, the 14.71 mg/kg Cd treatment groups were significantly lower than non-irrigated soil, indicating that long-term high Cd stress caused greater oxidative damage to *B. papyrifera* seedlings. It could also be seen from the change of time that the accumulation of Cd led to the increase in MDA content and the activation of antioxidant enzyme activities in *B. papyrifera* seedlings (Figure 4A–D).

#### 2.5.3. Changes in Chlorophyll Content

The chlorophyll content in plants is closely related to photosynthesis and nutritional status, and is an important indicator reflecting the growth of plants. The chlorophyll content (chlorophyll *a*, chlorophyll *b* and total chlorophyll) of the *B. papyrifera* leaves showed an overall upward trend with the extension of Cd stress time (Figure 5). At 12 h, the chlorophyll content under 5.71 mg/kg and 8.28 mg/kg Cd concentrations was significantly lower than that of non-irrigated soil, while the difference between high Cd concentrations (11.49 mg/kg and 14.71 mg/kg) and non-irrigated soil was not significant (*p* > 0.05). As time extended, the difference between 5.07 mg/kg, 5.71 mg/kg and 8.28 mg/kg decreased. On the whole, the content of chlorophyll in non-irrigated soil was higher than that of other Cd treatment groups. Furthermore, at 30 d, there was a significant difference (*p* < 0.05); the lowest chlorophyll content was at 11.49 mg/kg Cd concentration. The comparison of chlorophyll *a* and chlorophyll *b* content found that chlorophyll *b* content was higher than chlorophyll *a* under the same treatment conditions. The results indicated that Cd had an inhibitory effect on chlorophyll synthesis in the *B. papyrifera* seedlings.

## 3. Discussion

Elevated concentrations of essential and non-essential heavy metals in plants can inhibit plant growth and cause toxic symptoms. However, plants have a range of potential mechanisms at the cellular level that may be involved in detoxification, and thus resistance to heavy metal stress. For example, organic solutes such as soluble protein, soluble sugar, proline and organic acid are synthesized in the cell to reduce the intracellular osmotic potential and maintain the normal supply of water. The efflux of heavy metals reduces the absorption of heavy metals on the plasma membrane. In addition, plants could increase their enzyme activity to drive off reactive oxygen species (ROS), thereby repairing stress-damaged proteins [44,45]. Under normal circumstances, plant intake of large amounts of heavy metals may stimulate the formation of ROS in the body. These ROS, especially H_2_O_2_, are the most influential signal transduction factors in plant growth [46,47], which can cause oxidative stress reaction with proteins, lipids and other substances in plants [48]. In this process, the cell membrane permeability is enhanced, there is a lack of necessary metal in metalloprotein complexes, and toxic heavy metals are eventually replaced [49], while the removal of ROS is an important measure to reduce the damage to biological macromolecules such as proteins, sugars and nucleic acids. The balance between antioxidants and free radicals can play a useful biological function without causing too much damage [50].

Plant biomass has been widely used to evaluate the Cd tolerance of plants. Studies have shown that the plant biomass decreases significantly when the Cd content in soil is high [51,52]. In this study, Cd had a greater impact on the biomass of *B. papyrifera* root. When the Cd concentration reached 8.28 mg/kg, the biomass of the root decreased significantly compared with non-irrigated soil (5.07 mg/kg) (*p* < 0.05), while the biomass of the leaf and stem was not affected by Cd. This shows that Cd seriously affects the growth of roots. At the same time, under the high Cd concentration (14.71 mg/kg), the Cd content in the root of *B. papyrifera* was the highest, which was about three times and twice that in the stem and leaf, respectively. The research on Cd content showed that at low Cd concentrations (5.07 mg/kg and 5.71 mg/kg), Cd was mostly accumulated in the leaf, while it was mainly accumulated in the root at high Cd concentration. This is different from the accumulation of Mn in *B. papyrifera* [22]. The BCF of *B. papyrifera* increased significantly at the Cd concentrations of 11.49–14.71 mg/kg, while TF decreased significantly at 8.28–14.71 mg/kg. The above results indicated that *B. papyrifera* has a strong absorption capacity for heavy metals in the soil under high Cd concentration, and the metals mainly accumulate in the underground part; this may occur because the root cell wall of *B. papyrifera* selectively absorbs and fixes Cd in the soil, so most of the Cd was trapped in the root [23,53]. In addition, most studies have also shown that heavy metals can reduce the MC of plants [54,55]. In our research, there was no significant difference in the MC of each part of *B. papyrifera*, indicating that the tissues of *B. papyrifera* have stronger water holding capacity.

As an important osmotic adjustment substance and nutrient, soluble protein can increase the water retention capacity of the cells and protect the biofilm. In our research, the soluble protein content of *B. papyrifera* leaves decreased with the increase in Cd concentration at 12 h. At this time, the increase in MDA content indicated that it may be caused by the accumulation of ROS in the body [54,56,57]. MDA content is an important indicator reflecting the strength of lipid peroxidation, which indirectly reflects the severity of cells attacked by ROS [58]. With the prolongation of time, the soluble protein content increased significantly in order to adapt to the adversity. The *B. papyrifera* maintained the normal metabolism of the cells and increased the stress resistance by increasing the number of functional proteins [59]. From the results, we can see that in the non-irrigated soil (Figure 3), the soluble protein content first decreased and then increased with the extension of Cd treatment time. According to the test results of the Cd content in the soil, this may be related to the Cd contained in the soil itself. Moreover, the soluble protein content of *B. papyrifera* seedlings in the Cd treatment groups also increased significantly at 30 d, indicating that the *B. papyrifera* seedlings could detoxify by increasing the protein content after adapting to the Cd stress environment.

It can be seen from the change of soluble sugar content that with the increase in Cd concentration and the extension of stress time, *B. papyrifera* adjusted the osmotic pressure to adapt to environmental pressure by increasing the soluble sugar content. Soluble sugars in plants, such as sucrose, glucose and fructose, can offer protection from freezing damage caused by low temperature stress and can induce an increase in enzyme activity to drive out higher H_2_O_2_; for this reason, they are known as good membrane protectants [60,61,62]. Moreover, transcriptome analysis confirmed that gene expression involved in the regulation of sugar signal transduction is associated with oxidative stress [63]. This is why the *B. papyrifera* has an increase in soluble sugar at high Cd concentration.

Under stress conditions, proline accumulated in plants not only acts as an osmotic regulator in plant cytoplasm, but also plays an important role in stabilizing the structure of biological macromolecules and regulating the redox potential of cells [64,65]. Our research found that the proline content in all the experimental groups was in an increasing trend due to the influence of the Cd content in soil itself and the application of Cd solution. Studies have shown that heavy metals such as Cd, Cu and Zn can induce the production of proline in plants [66,67,68]. Proline plays a protective role in heavy metal toxicity by inhibiting lipid peroxidation [67]. This was related to the water balance of plant leaves [68]. Chen et al. [69] found that abscisic acid (ABA) induced an increase in proline content in rice under Cu stress, and was related to the increase in amino acids. It was subsequently confirmed that amino acids and phytochelatins played a role in metal binding, and participated in the antioxidant process, promoting the increase in proline content [70]. Whether the increase in proline content in *B. papyrifera* is related to Cd induction and the regulatory mechanism remains to be confirmed.

The determination of MDA content is often combined with the determination of the activity of antioxidant enzymes. SOD, POD and CAT are key enzymes in the antioxidant system [71,72]. Under 5.71 mg/kg Cd concentration, the content of MDA in *B. papyrifera* decreased, indicating that *B. papyrifera* could adapt to low Cd stress. High Cd concentrations (11.49 mg/kg and 14.71 mg/kg) could accumulate ROS in *B. papyrifera*, leading to an increase in MDA content. Our study found that SOD, POD and CAT played a role in scavenging ROS, but SOD and POD activities were significantly higher than CAT. This phenomenon has been reported previously [22,23,58], and it was indicated that SOD and POD played a major defensive role in preventing oxidative damage under Cd stress. CAT can participate in the catalysis of H_2_O_2_ decomposition and reduce H_2_O_2_ content under stress [73]. The MDA content also showed a significant increasing trend in the non-irrigated soil as time progressed, indicating that the *B. papyrifera* suffered oxidative damage. At the same time, the activity of antioxidant enzymes increased, which protects cells from Cd poisoning by removing ROS.

Cd can inhibit the activity of chlorophyll biosynthetic enzymes, which in turn affects the function of chloroplasts [74]. This is consistent with our results, that is, the chlorophyll content under other Cd concentrations was lower than that of the non-irrigated soil. Cd has an effect on the photosynthetic system of *B. papyrifera* seedlings by inhibiting the activity of chlorophyll biosynthetic enzymes. In a short period, the effect of low Cd concentrations (5.71 mg/kg and 8.28 mg/kg) on chlorophyll content was greater than that of high Cd concentrations, which corresponds to the chloroplast content of palisade tissue in anatomic structure of leaves. With the extension of time, the adaptability of *B. papyrifera* to the stress environment increased, so the chlorophyll content increased rapidly at low Cd concentrations. Studies have shown that when plants were exposed to lower Cd concentrations, their chlorophyll content was higher than that of the control [54,75]. We believe that low Cd concentration could inhibit chlorophyll synthesis in *B. papyrifera* leaves. In addition, we found that Cd changed the content of chlorophyll type in *B. papyrifera* leaves. As heliophytes, the ratio of chlorophyll *a/b* should be larger, namely chlorophyll *a* > chlorophyll *b* [76]; the results of this study proved that the content of chlorophyll *b* was higher than chlorophyll *a*, which is also contrary to the chlorophyll synthesis of *B. papyrifera* under Mn stress [22]. Hence, the changes in chlorophyll *a/b* may be due to the special role of Cd in *B. papyrifera* chlorophyll synthesis.

Cadmium is toxic to plants even at very low concentrations [77]. In pot experiments, the toxic effects of Cd are more complicated, because Cd and other elements have interacted [78]. The background value of the soil in our experiment was much higher than in other studies [79,80], and the Cd content of the non-irrigated soil group has exceeded the threshold requirement for growing crops [81,82]. The above defects are due to the purchase of nutrient soil that does not meet the expectations of the experiment through a commercial company. The lack of a blank control prevented us from exploring the response mechanism of *B. papyrifera* to Cd stress. We could, however, still explore some details about morphological and physiological changes of *B. papyrifera* seedlings in Cd contaminated soil and confirm that *B. papyrifera* has good resistance to Cd stress. Optimistically, there are some details in the experiment that effectively slowed down the influence of the background value. First, before the stress experiment started, the *B. papyrifera* seedlings were adapted to the nutrient soil for two months and the influence of the background Cd could have been almost negligible. Secondly, as a woody plant, *B. papyrifera* is much more resistant to Cd concentration than herbs. This is also the source of our motivation to look for woody restoration plants and explore their mechanisms. On the whole, the research still showes that *B. papyrifera* reveals good Cd tolerance and has potential of its use as a phytoremediation.

## 4. Materials and Methods

### 4.1. Source of Seedlings and Cd Treatment

The plant material was an annual *B. papyrifera* cutting seedling, which was provided by Hunan Yuanyufeng Agricultural Co., Ltd. (Chenzhou, China). The *B. papyrifera* seedlings were planted in plastic pots with an outer diameter of 21 cm and a height of 12 cm, with 700 g soil not administered with a Cd solution (non-irrigated soil). The pot was kept in the greenhouse at 27 °C with a 14 h photoperiod and 50–60% of relative air humidity. In order to prevent the loss of Cd solution, a tray was padded on the bottom of the pot. After two months of adaptive training, 15 pots of seedlings with the same growth and health status were selected for Cd stress treatment. The average height of *B. papyrifera* seedlings was about 40 cm (*p* > 0.05).

During the Cd stress treatment, 400 mL CdCl_2_•2^1/2^H_2_O solutions with Cd concentrations of 10, 50, 100 and 150 μmol/L were uniformly injected into the soil of each pot. Non-irrigated soil was watered with the same volume of deionised water but without added Cd. Then, the plants were subjected to five levels of soil contamination corresponding to 5.07, 5.71, 8.28, 11.49 and 14.71 mg Cd/kg soil, of which the 5.07 mg/kg Cd was detected in the non-irrigated soil. Each treatment concentration was set to three replicates. All treated *B. papyrifera* seedlings were carried out in the greenhouse, and were maintained by regularly pouring an equal amount of deionized water into the trays. After 180 d treatment, the trees were harvested.

### 4.2. Determination of Growth Indicator and Cd Accumulation

The height of each experimental plant treated with different Cd concentrations was measured at 0 d, 90 d and 180 d, respectively. After 180 d of Cd stress treatment, the whole plants were harvested and washed with deionized water, then were divided into roots, stems and leaves. The fresh weight (FW) of roots and stems was measured. Then, they were dried at 105 °C for 1 h, and at 65 °C to constant weight and measured dry weight (DW). The moisture content (MC) of individual plant roots and stems was calculated by: MC (%) = (FW − DW)/FW ∗ 100. The proportion of root biomass in the whole plant was based on the following formula: DWroot/(DWroot + DWstem + DWleaf) ∗ 100%.

The dried plant tissues were ground into powder by a pulverizer (Yongkang Boou Hardware Products Co., Ltd., Yongkang, China, Dongyi: 200 T). Quantities of 0.5000 g of the pulverized plant samples were accurately weighed and placed into 100 mL erlenmeyer flasks; 5 mL of HNO_3_ was added, and the bent-neck funnel was covered, shaken well, and left to stand overnight. The next day, they were placed on an electric hot plate and heated in a fume hood at 160 °C for 1 h, 170 °C for 1 h, 180 °C for 1 h, and 190 °C for 30 min until the brown gas was almost exhausted. The erlenmeyer flasks were taken off and cooled, 1.5 mL HClO_4_ was added, and then the temperature on the electric heating plate was again raised to 200 °C and heated for 1–2 h to produce thick white smoke. At this time, most of the HClO_4_ was volatilized. When the solution in the erlenmeyer flasks was colorless and transparent, and the plant residue was white, the erlenmeyer flasks were taken off and cooled.

In addition, the soils were air-dried naturally, crushed in a mortar, and passed through a 100 mesh nylon sieve. Quantities of 0.2000 g of the soil samples were accurately weighed and placed into a 100 mL polytetrafluoroethylene (PTFE) crucible. The fume hood was turned on, and 10 mL of HCl was added to the PTFE crucible. It was shaken gently, placed on a hot plate, and heated at 120 °C for about 1 h. When the solution was fast-drying (3 mL), it was removed and cooled. Then, 5 mL HNO_3_, 5 mL HF and 3 mL HClO_4_ were added, and the solution was covered and heated at 185 °C for about 2 h; the lid was opened to drive off the white smoke, and when about 3 mL remained, the PTFE crucible was removed for cooling to occur.

The plant samples and soil samples treated above were filtered with ultrapure water in 25 mL volumetric flasks, and the Cd content in the samples was measured with a flame atomic absorption spectrophotometer.

The calculation formula of Cd content in plant tissue was as follows:Cd content (mg/kg DW) = detected Cd concentration ∗ 25 ∗ 10^−3^/0.5 ∗ 10^−3^.

The calculation formula of Cd content in soil was as follows:Cd content (mg/kg DW) = detected Cd concentration ∗ 25 ∗ 10^−3^/0.2 ∗ 10^−3^.

The larger the bioconcentration factor (BCF), the stronger the ability of plants to accumulate Cd; the larger the translocation factor (TF), the stronger the ability of plants to transport Cd from the roots to the aboveground part. BCF = the content of Cd in plants/the content of Cd in soil; TF = Cd content in the aboveground part (leaf and stem) of the plant/Cd content in the underground part (root) of the plant.

### 4.3. Determination of Physiological Characteristics

Fresh leaves were collected after *B. papyrifera* was treated for 12 h, 10 d, 20 d and 30 d, respectively. Then, under ice water bath conditions, liquid nitrogen was added and the leaf was rapidly ground into 10% tissue homogenate. After centrifugation at 4500 rpm for 10 min, the supernatant was taken for determination of soluble protein content, soluble sugar content, proline content, MDA content, superoxide dismutase (SOD) activity, peroxidase (POD) activity, catalase (CAT) activity and chlorophyll content (chlorophyll *a*, chlorophyll *b* and total chlorophyll). The redundant supernatant was stored in a freezer at −80 °C. All of the above relevant indicators were tested by corresponding kits, which were purchased from Nanjing Jiancheng Bioengineering Institute (http://www.njjcbio.com/) and the specific operation methods were consistent with the manufacturer’s instructions.

### 4.4. The Observation of Leaf Anatomy

After two months of Cd stress treatment, fresh *B. papyrifera* leaves were selected and paraffin section technique [83] was used to observe changes in leaf anatomy. During the whole experimental operation, the cut leaves were fixed with FAA fixative which was prepared with 70% alcohol, glacial acetic acid and formalin. After 24 h, the materials were dehydrated with alcohol from low to high concentration. The concentration and time were set to: 85% (1 h), 95% (1 h), 100% (1 h) and 100% (1 h). In order to enhance the refractive index of the tissue, the materials were cleared with xylene. The concentration ratio and time were set to: 1/2 xylene + 1/2 alcohol (1 h), xylene (1 h), xylene (1 h), 1/2 xylene + 1/2 paraffin (30 min, repeated three times). The materials were taken out and processed in the constant temperature incubator (model: 303-00, purchased from Guangzhou Kangheng Instrument Co., Ltd., Guangzhou, China) by low temperature waxing (36 °C, 1 h) and high temperature waxing (55–60 °C, 2 h, repeated three times). After the materials had been immersed in wax for a sufficient period of time, it was necessary to embed the wax block, and then take it out after solidification and cut it into a suitable shape, and fix it onto a small wooden block with wax water. The wax tapes were cut to have a thickness of about 8 μm using a KYD-QP biological tissue slice machine (Hubei Kangyida Medical Technology Co., Ltd., Xiaogan, China), were adhered to the glass slides, and placed on the KYD-TK biological tissue booth machine (Hubei Kangyida Medical Technology Co., Ltd.) at 30 °C to 35 °C. After the wax tapes were flattened and the water was completely evaporated, the slides could be saved in the slide box. Subsequently, in order to make different structures within the cell tissue present with different colors and strong enough differences for observation, we de-waxed the slides with xylene and alcohol, double stained them with 1% saffron + 0.5% fast green, and the slides were sealed with a neutral gum and observed under the electron microscope.

### 4.5. Data Analysis

Graph analysis were performed using SigmaPlot 12.5 software. The Kolmogorov–Smirnov Test was used to examine the normality of the data distribution. SPSS 20.0 software was used for two-way analyses of variance (ANOVA), the least significant difference (LSD) test was performed to assess the homogeneity of variance (*p* > 0.05), and the Games–Howell test was used to test significant difference (*p* < 0.05). All measurements were repeated three times and the data were expressed as mean values ± standard error (SE).

## 5. Conclusions

In the research, we found that *B. papyrifera* adapts to damage caused by Cd stress via physiological and biochemical processes. *B. papyrifera* has a high biomass, even if Cd stress caused its root biomass to decrease significantly, and the MC of roots and stems were not affected by Cd stress. The roots of *B. papyrifera* could accumulate more heavy metals under high Cd concentrations, thus reducing the transfer to the aboveground part. The main detoxification mechanisms such as changes in the activities of antioxidant enzymes (SOD and POD) and the co-feedback of ROS played a major regulatory role. The increased protein of *B. papyrifera* seedlings and the developed root system constituted the primary barrier against external aggression. Cd could change the chlorophyll type and content of *B. papyrifera* leaves, and had a great influence on the synthesis of chloroplasts. In conclusion, our research has initially proved that *B. papyrifera* has the advantages of fast growth, a well-developed root system, strong soil and water conservation ability, and high biomass. *B. papyrifera* has strong enrichment ability under high Cd concentrations, making it suitable for being widely planted in heavy metal contaminated soil and used for soil and ecological environment restoration.

## Figures and Tables

**Figure 1 plants-09-01698-f001:**
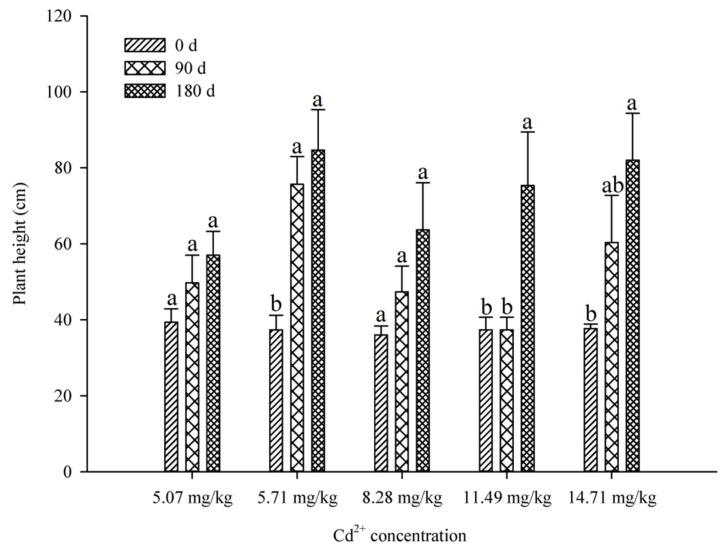
The height of *B. papyrifera* seedlings at soils of different Cd concentration. In this graph, 0 d is the height before Cd stress treatment; 90 d is the height of the *B. papyrifera* seedlings entering the stagnant growth period during winter; 180 d is the height of harvest. Results are expressed as mean ± SE of three replicates. Different letters indicate significant differences between the same indicators (*p* < 0.05).

**Figure 2 plants-09-01698-f002:**
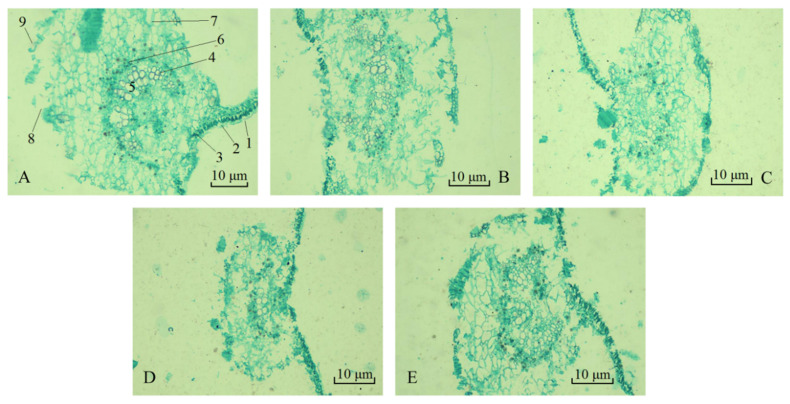
The anatomical structure of the transverse section of the *B. papyrifera* leaves. (**A**–**E**) (100×) correspond to 5.07 mg/kg, 5.71 mg/kg, 8.28 mg/kg, 11.49 mg/kg and 14.71 mg/kg of Cd treated *B. papyrifera* leaves, respectively. The numbers in (**A**) indicate: 1—cuticle, 2—upper epidermis, 3—palisade tissue, 4—xylem, 5—vessel, 6—phloem, 7—sponge tissue, 8—stomata, 9—lower epidermis.

**Figure 3 plants-09-01698-f003:**
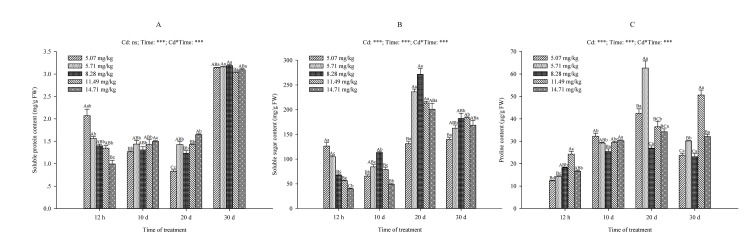
Effects of different Cd concentrations on the soluble protein content (mg/g FW) (**A**), soluble sugar content (mg/g FW) (**B**) and proline content (μg/g FW) (**C**) of *B. papyrifera* seedlings. Results are expressed as mean ± SE of three replicates. Different uppercase letters indicate significant differences (*p* < 0.05) between Cd content, and different lowercase letters indicate significant differences (*p* < 0.05) between treatment times. Cd, Cd content effect; Time, time effect; Cd*Time, the interactive effect of Cd content and time. ns, not significant; *** represents *p* ≤ 0.001.

**Figure 4 plants-09-01698-f004:**
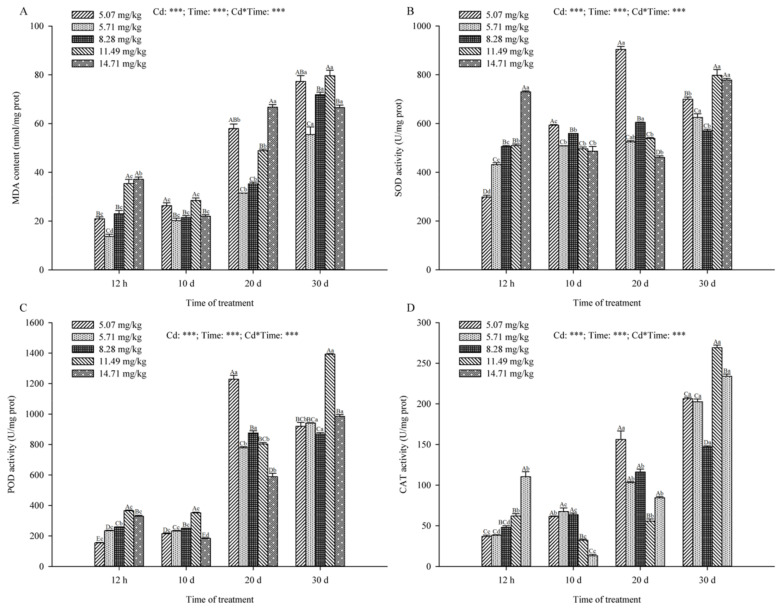
The effects of different Cd treatment concentrations on the MDA content (nmol/mg prot) (**A**), SOD activity (U/mg prot) (**B**), POD activity (U/mg prot) (**C**) and CAT activity (U/mg prot) (**D**) of *B. papyrifera* seedlings. Results are expressed as mean ± SE of three replicates. Different uppercase letters indicate significant differences (*p* < 0.05) between Cd content, and different lowercase letters indicate significant differences (*p* < 0.05) between treatment times. Cd, Cd content effect; Time, time effect; Cd*Time, the interactive effect of Cd content and time. *** represents *p* ≤ 0.001.

**Figure 5 plants-09-01698-f005:**
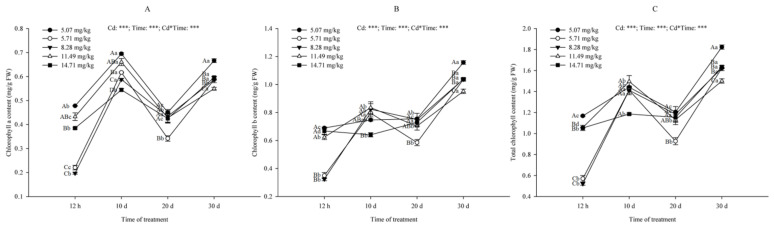
Effects of different Cd treatment concentrations on chlorophyll *a* content (mg/g FW) (**A**), chlorophyll *b* content (mg/g FW) (**B**) and total chlorophyll content (mg/g FW) (**C**) of *B. papyrifera* seedlings. Results are expressed as mean ± SE of three replicates. Different uppercase letters indicate significant differences (*p* < 0.05) between Cd content, and different lowercase letters indicate significant differences (*p* < 0.05) between treatment times. Cd, Cd content effect; Time, time effect; Cd*Time, the interactive effect of Cd content and time. *** represents *p* ≤ 0.001.

**Table 1 plants-09-01698-t001:** The dry weight (DW), fresh weight (FW) and moisture content (MC) of root, stem and leaf for the *B. papyrifera* seedlings at different Cd concentration soils. The data are presented with the mean ± SE of three replicates. Different letters indicate significant differences between the same indicators (*p* < 0.05).

Cd Concentrations (mg/kg)	Root			Stem			Leaf ^†^
	FW (g)	DW (g)	MC (%)	FW (g)	DW (g)	MC (%)	DW (g)
5.07	28.72 ± 2.84 a	6.30 ± 0.57 a	78.02 ± 0.00 a	13.65 ± 1.50 a	4.91 ± 0.21 a	63.00 ± 0.09 a	5.32 ± 0.57 a
5.71	16.51 ± 0.45 b	3.85 ± 0.14 ab	76.70 ± 0.01 a	15.89 ± 3.47 a	6.01 ± 1.29 a	62.05 ± 0.02 a	6.00 ± 0.80 a
8.28	13.20 ± 0.56 b	3.02 ± 0.22 b	77.03 ± 0.04 a	11.39 ± 2.04 a	4.48 ± 0.81 a	60.45 ± 0.04 a	4.52 ± 1.00 a
11.49	17.56 ± 3.37 b	3.94 ± 0.81 b	77.74 ± 0.02 a	12.22 ± 0.90 a	4.15 ± 0.27 a	66.00 ± 0.01 a	4.28 ± 0.28 a
14.71	16.33 ± 2.12 b	3.53 ± 0.51 b	78.50 ± 0.01 a	11.46 ± 1.61 a	4.37 ± 0.73 a	62.14 ± 0.03 a	3.87 ± 0.82 a

† Note: The determination of leaf FW and MC was omitted due to deciduous phenomena in the *B. papyrifera* seedlings, which occurred as a result of seasonal reasons and the long experiment period.

**Table 2 plants-09-01698-t002:** Cd content in the root, stem and leaf of *B. papyrifera* and the accumulation and translocation of Cd in *B. papyrifera* after 180 d of Cd treatment. The data are presented with the mean ± SE of three replicates. Different letters indicate significant differences between the same indicators (*p* < 0.05).

Cd Concentrations (mg/kg)	Cd Concentration (mg/kg DW)			BCF	TF
	Root	Stem	Leaf		
5.07	0.15 ± 0.04 b	0.60 ± 0.10 bc	1.20 ± 0.10 c	0.12 ± 0.01 b	6.87 ± 1.74 a
5.71	0.26 ± 0.10 b	0.41 ± 0.04 c	1.43 ± 0.15 bc	0.14 ± 0.00 b	4.81 ± 1.88 a
8.28	1.58 ± 0.28 ab	0.68 ± 0.07 b	1.65 ± 0.14 b	0.18 ± 0.02 ab	0.78 ± 0.14 b
11.49	1.43 ± 0.04 ab	1.31 ± 0.04 a	1.46 ± 0.07 bc	0.19 ± 0.01 a	0.97 ± 0.04 b
14.71	4.28 ± 1.69 a	1.43 ± 0.04 a	2.10 ± 0.04 a	0.21 ± 0.02 a	0.56 ± 0.21 b

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
