# Peer review of "Morphological and Physiological Changes of Broussonetia papyrifera Seedlings in Cadmium Contaminated Soil"

_plants, 2020, doi:10.3390/plants9121698_

Round 1
Reviewer 1 Report
Authors performed revision but some important points which are raised previously did not receive sufficient attention and explanations are unclear for me. WITH ALL RESPECT, SOME DATA ARE CLEARLY WRONG, THIS IS NOT ACCEPTABLE SCIENCE EVEN IN PAID OPEN ACCESS JOURNAL.
Only main comments:
i) protein content as g prot/L FW is wrong, read millions of papers where proteins are expressed per unit of weight, i.e. mg/g FW or DW. Or do you expect that plant you studied has organs in liquid state? Only then expression per L is OK.
ii) sugar content as ug sugars/mg protein FW is nonsense and is not explained correctly, I would like to see kit which recommends such calculation. The link you provided is in Chinese language and I am not able to read it.
iii) chrorophyll content ca. 0.6 mg/g indicates that plants did not receive sufficient illumination and I do not see their photo (link to supplementary materials does not work), leading to unclear microscopic photos (scale bar is missing, I assume various magnification of individual photos!).
iv)POD and CAT activities seem to be overestimated at first sight
Overall, work has weak experimental design (too high/environmentally irrelevant Cd doses) and contains many technical problems, indicating that authors are not biochemical/physiological experts and were not able to consult problems with experts when preparing revision.
Reviewer 2 Report
REVIEWER’S DECISION:
The article might be acceptable for publication, however there are several issues that must be taken into consideration, amended, improved or at least discussed. Due to the relatively high number of questions and problems indicated, I qualify the work as requiring major revision, but I hope this time the authors can manage to correct the paper quite easily without a need to perform additional complementary experimenting.
____________________________
GENERAL COMMENTS
The paper presents original research which fits the aims and scope of the Plants special issue dedicated to abiotic stress. Among the problems requiring corrections or explanations are misleading treatment concentration data, units, gaps in interpretations as well as other minor mistakes, as listed below. Compared to the previous version, I admit that the authors have made much effort to improve the content, including the acceptable response to my previous remarks. Also the English language has been corrected – it has obviously been improved. However, there are still numerous problems and therefore I suggest, one more time, having the text checked and improved by a professional English editing office, for example making use of the commercial opportunity given by the MDPI Editorial.
I believe that the presented data are worth publishing despite some criticism pronounced by other reviewers, regarding the paper as not bringing striking novelties. Cd is still a serious environmental problem and it is reasonable to search for new phytoremediators as well as to describe them. In addition, the studied model exhibits interesting physiological mechanisms in term of the response to excessive Cd.
Again, it has to be emphasized that the authors have considerable experience and achievements in several fields related to ecology and environmental issues including microbial remediation and phytoremediation of heavy metals. They have also developed sound methodological expertise which makes the presented data reliable.
This paper describes Broussonetia papyrifera as an efficient phytoremediation model plant revealing interesting response mechanisms towards Cd treatment. Some unique features of B. papyrifera as revealed in the work I consider as a significant contribution to the general knowledge on plant-abiotic stress reactions as well as to the application practice where the studied plant might be used for Cd contamination phytoremediation.
The experimental work has been carried out, in general, properly, is well documented, analyzed, presented and statistically evaluated.
All the Figs and tables including Supplementary material are necessary and relevant to the content, and give direct data support for the Results. Discussion is complete and relevant.
Questions and issues that should be included or amended in the revised paper are listed below.
________________________________
The main issues that require corrections or explanations:
(1) English has been evidently checked and is now much better, except for some specific fragments where it still requires modifications, which is sometimes essential. There are still problems with singular/plural use, articles (like “the” ahead of names, in many cases the articles are lacking), sequence of tenses, style etc., which make me suggest the language check as emphasized above.
(2) PROBLEMS WITH Cd CONCENTRATIONS USED FOR TREATMENT. The authors inform that the concentrations of Cd were applied at [μmol/L] but the whole experimental setup was arranged as a pot test. So – such information is almost useless or at least misleading; it has to be precisely defined what was the real Cd concentration – expressed as a dose per pot or mg Cd per kg soil d.w. (indeed such data are then stated, e.g. in Table S1, but it has to be described explicitly, I suggest in Materials and Methods – in a chapter on Cd treatment). Note that one cannot mix these units, which may bring severe difficulties in understanding the experimental and physiological status of Cd-treated plants. Please, make sure you give the precise, crystal clear, unequivocally described information, since it is crucial for right understanding of the work, details are required on the way you treated the plant seedlings – not only by referring to the earlier published work. In addition to the problem of realizing actual (real) levels of Cd interacting with plant seedlings (that is the amount of Cd in soil, not in the solution used for irrigation), the current description makes it extremely difficult to discuss the plant’s response towards Cd in the context of other, numerous literature sources regarding other Cd-treated plants.
So, were these concentrations used for watering/irrigating the plants systematically or just treated once? What was the final Cd concn. in the soil determined upon treatment with each particular Cd solution? This problem occurs throughout the whole work, at first it appearas in Abstract. It also refers to the Methods chapter – “Cd treatment”. Some examples to show the problem and comments are given below:
ABSTRACT, line 17
different cadmium (Cd) conditions (0, 10, 50, 100 and 150 μmol/L), potted experiments...
Unclear and misleading, since the experiment was not a hydroponic culture but a pot test, rather. It is not clear whether the mentioned concentrations were real ones applied to treat the plants or were just the concentrations used to water (irrigate) the plants in pots? If so, the Cd levels should be recalculated as doses per pot, per kg of d.w. soil? Or given in other way so as to give straightforward information on the real treatment level. In other words: what was the real Cd concentration that the plant had to deal with?
lines 17,18
The results showed that Cd mainly concentrated in the root when the concentration of Cd was high (150 μmol/L),
or in the Discussion:
When the concentration reached 50 μmol/L ?
where was such a concentration reached? It was applied rather than reached in the soil of the pot used for testing!
AND SO ON... the same problem refers to Figures, tables, etc.
(3) NEED TO ENRICH THE DISCUSSION slightly: The data show very complex dependencies. The authors, while discussing the results, try to ascribe or link certain parameters to the physiological state of the plant responding to different Cd levels. Figs 3 and 4 as well as Supplementary 2 and 3 show numerous parameter values (soluble protein, sugar, Pro, MDA, antioxidant enzyme activities...) as dependent on the Cd treatment variants. However, it has to be noticed that in the CK control the plants showed significant parameter changes, as well. Why is it so? Expected or unusual? What caused such a variability? It is therefore of high interest to discuss all the observed changes in the context of the CK variability – the latter was left unexplained.
(4) Abbreviation list – please add “CK” to the abbreviation list, for clarity and remove obvious symbols such as element symbols (As, Mn, Cd, Zn, Fe, N) taught in primary schools.
(5) Also – please make sure the present way of expressing units is acceptable and easily understandable, my impression is that the units are currently difficult to read – suggest: you might add a space before [g prot] or [mg prot] for clarity, all throughout the text and Figures. If not – give convincing explanation for the applied formalism.
(6) Other still existent problems with English, misleading phrases, technical issues, mistakes to be corrected – line by line:
lines 17,18
and the root biomass was significantly reduced by Cd stress although it promotes the growth of seedlings.
-> Please rewrite the unclear phrase – correct the use of tenses (was – is?), what doest “it” refer to (was it cadmium or root biomass reduction)?
line 20
bioconcentration factors (BCF) increased with the increase of concentration, and reach the
-> increase of what concentration? cadmium?
-> and “reached” not reach
line 23
However, B. papyrifera can synthesize organic solutes
-> However, B. papyrifera could synthesize organic solutes
Abstract, e.g. line 25
Cd stress and is a good tree species for soil and ecological environment restoration...
-> overuse of colloquial “good” – it is not a formal English, sounds too “loose”, 3 times used in Abstract, please use appropriate “language register” and more elegant vocabulary, scientific formal style.
line 34
environment soil
-> soil environment
by replacing the soil contaminated heavy metals is high costly
-> by replacing the soil contaminated WITH heavy metals is HIGHLY EXPENSIVE or COSTLY
line 37
basing on
-> based on
line 49
The morphological change of plants is the most intuitive response to abiotic stress.
suggested: changes in plants morphology ....
line 58
and the increase of antioxidant enzyme activity
-> and the increase of antioxidant enzyme activities (there are many antioxidant enzymes)
line 58
while high concentration
please define what concentration – of heavy metals or Cd and Mn?
line 67
The leaves are rich in amino acids, proteins, fats and trace elements, which propose it as a good source of feed
-> must correct – the leaves cannot “propose it as a good source”...
2.1. Height changes and the characteristics of leaf phenotypic
-> 2.1. Height changes and the characteristics of leaf phenotype or leaf phenotypic features
line 131
and all reaches the maximum
-> and all reached the maximum
line 140
Why “transportation”? instead of suggested: “translocation”
Please, also, support the caption with mentioning the respective parameters, so
instead of:
accumulation and transportation of Cd in B. papyrifera
rather suggested:
-> accumulation and translocation of Cd in B. papyrifera, as evaluated based on BCF and TF parameters, respectively
line 144
treated with different concentrations were observed
-> treated with different Cd concentrations were observed
line 184
which was 189.96 μg/mgprot.
shouldn’t the unit contain FW, that is: which was 189.96 μg/mgprot. FW
Discussion
line 245
The following sentence has no predicate – must rewrite and complement to make sense
In addition, by increasing the enzyme activity to drive off reactive oxygen species (ROS), thereby repairing stress-damaged proteins [44,45].
line 249
which can occur oxidative stress
-> wrong word used
“which can cause / which can lead to oxidative stress” ??
line 262 (and later)
Substitute the unfortunate term used “concentrated” for describing Cd tissue distribution; suggest “was accumulated”; “was found” “tended to cumulate”...
Cd was mostly concentrated in the leaf, while it was mainly concentrated in root at high concentration (??)
line 282
The change of soluble sugar content was reflected in the low Cd concentration, which can adapt to the environmental pressure by changing the osmotic pressure
-> must rewrite, the whole sentence makes no sense: sugar content reflected in concentration??
low Cd concentration can adapt? or: sugar content can adapt?
line 299
phytochelins??
-> phytochelatins ?
line 305
will accumulate?
why in future tense?
line 314
short period of time,
can a “period” be not related to time?
which was corresponds
must correct mistake
line 325
The following sentence has no subject and makes no sense:
(??) Prove the special role of Cd on B. papyrifera chlorophyll synthesis.
Materials Methods
4.2. Detection of nutrient soils
-> 4.2. Detection of nutrients in soils
l. 340 seedlings
l. 354 stirred
line 357
Each experimental plant height with different treated concentrations was measured at 0 d, 90 d
-> Height of each experimental plant treated with different Cd concentrations was measured at 0 d, 90 d
l. 360
Then dried them at 105 °C for 1 h, 65 °C to constant weight
->Then, they were dried at 105 °C for 1 h, 65 °C to constant weight
line 371
will be volatilized
-> why in future tense??
lines 385-387
In the formulae – calculations – when giving units, specify whether you used FW or DW – this should be included in the units definition
line 427
Graph analysis was performed using SigmaPlot 12.5 software
-> Graph analyses were performed using SigmaPlot 12.5 software
Conclusions
please watch for the right usage of tenses – avoid mixing past with present
line 444
It has strong enrichment ability under high Cd concentration,
Please rewrite – what is a “strong enrichment ability “ ? enrichment in what?
__________________________________
Round 2
Reviewer 1 Report
Authors improved method parameters (proteins, sugars) I commented but graphs are not unclear and I am not able to read units and parameters. I am not therefore able to comment improvements. Why leaf MC is not presented? If I see correctly from the graphs, e.g. Fig. 3, 4 and 5 represents data from treatments up to 30 days but Cd content (Table 2) was assayed after 180 days, why? Physiology must be essayed in the same day as Cd content. I am sure that 180 days of cultivation with only 0.7 kg soil per pot certainly led to Cd dillution by growing biomass, leading to erronous interpretation - you should add continually Cd during such long experiment. Alternatively, quantify Cd in the same day as physiology = after 30 days. Microscopic photos are still unclear and without scale bar.
Reviewer 2 Report
REVIEWER’S DECISION:
Although the paper has been generally satisfactorily improved in response to the reviewers’ suggestions, a new substantial problem has emerged and I regrettably have to conclude it still needs MAJOR REVISION due to the lack of CONTROL EXPERIMENTS. This time there appeared an important issue regarding the “control” which was in fact a severely polluted, although “non-irrigated” soil. Therefore, the objects depicted as a “control group” cannot serve as any control, at all. Previously, the authors just mentioned the irrigation solution applied to the soil but the information regarding the original Cd contamination and final Cd determination in the treated soil was lacking. This point has now been revealed by showing real Cd concentration levels in the soil, which I earlier indicated as a crucial lacking information. The authors now reliably and fairly show that their soil was originally affected by extreme Cd contamination (!) of 5 mg/kg. In terms of the current environmental regulations it is not legitimate to state that such a Cd level “reflects the widespread of Cd pollution in daily life”. This is not true. Such soil in field conditions SHOULD NOT BE USED as a cultivation substrate (=nutrient soil) for plants and should be subjected to remediation itself! The mentioned Cd level is close or even beyond permissible levels for agriculture soil, which means that the “control” seedlings respond to high abiotic stress and there is no information what would be the real condition of the Cd-UNTREATED seedlings. In consequence of the lack of the basic control experiment, the study cannot compare any changes in the seedlings physiology, chemistry and biochemistry, because there are no “reference values” for such comparisons.
GENERAL COMMENTS
The main issue
For control experiments the authors must PROVIDE ALL the DATA complemented parallelly with the appropriate seedlings grown in the soil LACKING CADMIUM, which is a fundamental requirement for ANY COMPARATIVE WORK. Any conclusion regarding observed changes in ANY PARAMETER VALUE is meaningless unless you show it compared to the control, this is an elementary necessary practice in papers dealing with the abiotic stress reaction.
To give just one representative, selected example, please see the recent paper of Yang Zhi et al. in Frontiers in Chemistry, “Mechanism of Remediation of Cadmium-Contaminated Soil With Low-Energy Plant Snapdragon” https://doi.org/10.3389/fchem.2020.00222. The authors treated the plant with the concentrations several-fold lower than your control soil! In their work: “the response of snapdragon in a pot-culture experiment under two concentrations of Cd (1.0 and 2.5 mg/kg) was evaluated. The authors also claim that:
“Cadmium (Cd) is a potentially harmful heavy metal that can be toxic to plants even at very low concentrations (0.5 μg Cd g−1 soil).”
Note that the 5mg/kg soil Cd level indeed has to be considered very high in the light of the current norms and permissible levels. For the relevant references see the data in:
Tóth, G.; Hermann, T.; Da Silva, M.R.; Montanarella, L. Heavy metals in agricultural soils of the European Union with implications for food safety. Environ. Int. 2016, 88, 299–309.
Adagunodo et al., Heavy metals' data in soils for agricultural activities. Data in Brief 18, 1847-1855, https://doi.org/10.1016/j.dib.2018.04.115
See also WHO levels of 1996.
To mention, the Dutch target Cd values for soils are given as 0.8 mg/kg, the threshold limits are 1 mg/kg and the permissible limits for the EU are 10 mg/kg, whereas in some countries these limits are even stricter, like e.g. in Poland, where they range from 2 to 5 mg/kg soil d.w., depending on the soil subgroup.
I have no idea why the authors used the highly contaminated soil for their experiments, which resulted in their experimental set-up involving the Cd-treated plants, only.
In addition, the highest Cd levels of treated plants were just about 15 mg/kg, so every result shown refers to the plants response to a comparable set of concentrations, with no control, again.
Moreover, how can you be sure of the identity of the Cd forms occurring originally in the soil and then in the “treated” samples – note that the first experimental point amounts to 5.71 mg/kg which is a concentration just about 13% higher than in the original “background” soil. I can assume that the authrs have not carried any Cd mobility (speciation, bioavailability) evaluation.
Other comments
In general, the authors’ comment on the subject and the chosen research model, that is their attempt to promote woody plants as efficient phytoremediators of heavy metal contamination sound convincing. I have emphasized this previously – that in my opinion the topic is worth publishing despite the fact that no profound changes were noted and described in terms of Broussonetia papyrifera response to the cadmium environmental stress. Still – these potential changes were observed with no control and in the relatively narrow range of Cd concentrations.
The experimental setup description changing the concentrations given to the new units that is mg/kg obviously yields a clearer picture – enables to realize the real Cd concentrations that the plant roots had to deal with. I admit that the [mol/L] irrigation still can be acceptable provided the more detailed description of the cultivation and treatment method but it is now clear that the irrigation just added some Cd to previously highly polluted soil.
It is very polite of the authors to mention the reviewers’ suggestions in the Acknowledgements, but please, bear in mind that, upon request to review the paper, I felt obliged to make any effort to improve the general message of the hard experimental work done. This is what I am doing now, as well.
The lack of control experiment possibly brings an answer to my previous Point 3. remark, to cite:
“...it has to be noticed that in the CK control the plants showed significant parameter changes, as well. Why is it so? Expected or unusual? What caused such a variability? It is therefore of high interest to discuss all the observed changes in the context of the CK variability – the latter was left unexplained.”
Obviously the CK plants were already responding to high indigenous Cd level as determined in the “non-irrigated” soil!
As the “control” experiment is non-existent, there is no “control group”, so, the authors cannot claim in the new version:
“Based on the above reason, we still measured the physiological parameters of the control group every time to ensure the control effect.”
Other remarks and minor points
I believe the English has been improved – at least my previous suggestions have all been considered and wherever possible, implemented and mistakes corrected. The message is now much straightforward, and I cannot be of any more help as myself I am not a native English speaker.
Among some minor faults that I noticed:
line 18
and the root biomass is significantly reduced by Cd stress although it promotes the growth of seedlings.
-> and the root biomass was significantly reduced by Cd stress although Cd promoted the growth of seedlings.
line 354
Then the plants were subject to five levels of soil contamination
->Then the plants were subjected to five levels of soil contamination
Round 3
Reviewer 2 Report
REVIEWER’S DECISION:
The paper can be now published after “MINOR REVISION” – it needs some additional explanations preferably in the Discussion. The authors can make use of some arguments provided in the “Response to the reviewer” – these seem convincing enough to explain the validity of the data obtained while using the Cd-contaminated soil as a non-irrigated substrate for plant cultivation. Please see the comments below.
GENERAL COMMENTS
The problems with high Cd content in a commercial soil substrate.
I understand the situation but the authors have to agree that it is their responsibility to select the appropriate materials only and apply the optimal methodology. I regret to learn that the soil produced by a source company has been so severely contaminated with Cd (and this is obviously an accepted “daily life” event! You cannot grow agricultural plants on such substrates).
Below are some of my doubts while taking a decision that the article in the present form might appear provided some additional straightforward explanations are included.
I can see the high amount of data collected regarding the applied model plant and still have an impression that Broussonetia papyrifera should find its place in the literature on Cd phytoremediation. I appreciate the great effort made by the authors to make the data publishable. Their recent approach involves substantial change of the presentation form, which is quite a smart solution in the context of the troublesome situation. This helps much in terms of publishing the results and still enables concluding the most important part which shows that B. papyrifera reveals good Cd tolerance and has potential of its use as a industrial plant in environmental projects. However, the weak part still remains, and the authors have to be aware that many researchers may strongly criticize such experimental setup lacking control.
I do not agree that one can employ statistics to somehow generalize an organism’s response to a heavy metal, and use variance analyses to “extract” the effect of factors (point #2). This is because any physiological response is absolutely non-linear, and cannot be anyhow approximated to the zero level, which brings back the necessity of the blank control. This is one-to-zero condition, and in your model – you are just analyzing variant levels of Cd and cannot judge what would happen if there were no Cd at all. Yet, this is not the most important matter, here, in this study. Your idea (now) is to show the growth and physiological potential within some range of the heavy metal concentrations. The preadaptation during seedlings growth for 2 months is quite convincing in the context of the above. Also the rest of your argumentation in pt.3 and partially in pt.4 appears rational.
On the other hand, note that in the Yang Zhi et al. (2020) paper the “contaminated” blank soil contained only 0.15 and not 5 mg/kg!
Taking into account all the above, my final suggestion is that the authors provide the readers with the fair description of the facts, and I think that most of the argumentation topics given in the “Response...” are ready to be included as they bring important explanations in terms of defending the model. I suggest adding the relevant paragraph in the Discussion section, a chapter suitable for “discussion of science itself”. Therefore, this time I marked “minor revision”, hoping that at this stage the paper is almost complete and acceptable – at least for the problems I indicated as a reviewer.
Finally, I agree with the authors’ conclusion that “reading the literature, ... the same mistakes may exist in the field of phytoremediation.” Yes, numerous examples can be found of these as well as many other problems and misinterpretations and that is why it is sometimes so difficult to draw general conclusions and to compare different models applied. But this means we have to put even more emphasis on making our picture clearer.
Other remarks and minor points
I suggest modifying the phrase in line 348 (sounds wrong in English):
"soil without Cd solution irrigated (non-irrigated soil)."
into:
soil not administered with a Cd solution (non-irrigated soil).
or:
soil not irrigated with a Cd solution (non-irrigated soil).
In lines 354–355
I still claim that you should not state that 5 mg/kg soil is a “daily life” situation. Please, again, read my previous comments where I gave some norms and permissible Cd levels. Even if there were some real situation of such soil pollution – in this particular case the soil was purchased and applied as an “optimal” plant growth substrate, so I would rewrite the risky statement:
“5.07 mg/kg Cd was detected in the non-irrigated soil, reflecting the widespread presence of Cd pollution in daily life.”
Author Response
Please see the attachment.

This manuscript is a resubmission of an earlier submission. The following is a list of the peer review reports and author responses from that submission.
Round 1
Reviewer 1 Report
The impact of Cd on plants has been studied in thousands of papers. The work therefore lacks novelty and originality. At the same time, monitored parameters are only simple and contain numerous mistakes, indicating that experienced supervisor (if any) did not check the manuscript carefully. I am sorry but this is rather very weak experiment for BSc study, not for scientific publication. I can´t imagine what I would cite from such paper.
Main comments:
i) you write about hyperaccumulators but you did not quantify Cd content, why?
ii) photo of experimental plants, to show the impact on phenotype, is needed. Soil composition is only partially mentioned. Light conditions are unknown.
iii) microscopic photos show damaged object, differences are hardly visible.
iv) fig. 1 does not contain statistics. Tab. 1 does not contain three columns for leaf.
v) mainly biochemical data are weak: protein content as g prot/L? sugars expressed per mg protein?? Proline and chlorophyll contents do not contain measured unit (g FW or DW?). However, chlorophyll content is underestimated in each case. On the other hand, enzymatic activities are overestimated at first sight.
Overall, only simple and doubtful data are presented. The work would be much better with additional parameters such as general antioxidants and specific metabolites, precise anatomy of stem and root tissue, assay of xylem/phloem exudates etc.
Reviewer 2 Report
REVIEWER’S DECISION:
THE PAPER REQUIRES thorough check and may be accepted after SUBSTANTIAL REVISION due to numerous faults. The main drawback regarding the content is the lack of the data regarding the accumulated pool of Cd in roots and shoots (seedlings) as well as the level of Cd in soil administered with this metal.
The authors, please, also have to take more care about formal issues and details and focus on the contribution, explicitly, the paper has been written in a hurry. Let the article be verified by a competent language-editing professional or at least a native speaker; in the present form the language level is unacceptable and the general message suffers from unclarity, brings misunderstandings or sometimes even mess.
Despite the problems mentioned, there is sound and important scientific novelty presented, although based on selected data, only. The paper presents original research which fits the aims and scope of the Plants special issue dedicated to abiotic stress.
Questions and issues that should be included or amended in the revised paper are listed below.
GENERAL COMMENTS
The authors have considerable experience and achievements in several fields related to ecology and environmental issues including microbial remediation and phytoremediation of heavy metals. This paper is a new one in the series of research studies involving Broussonetia papyrifera, which has been proposed as an efficient phytoremediation model plant.
Previously, this tree was studied with regard to the resistance to salt and drought stresses, selected heavy metals as well as remediation of Mn-contaminated soils.
In this study, the Cd tolerance and adaptation mechanisms have been studied and described based on some selected parameters assessed (plant seedling biomass, height, leaf anatomy, content of chlorophylls, soluble proteins, sugars, Pro and MDA, chosen enzyme activities). Such an arbitrary selection of experimental work is one of the weak points of this paper; for example, adding the lacking information on photosynthesis apparatus efficiency would be very informative – now the authors can merely hypothesize about photosynthesis parameters based on chlorophyll content or “developed palisade tissue” (see l. 131).
The work brings some novel information, although it is not revolutionary, as there have been numerous plants tolerant to Cd and proposed to be efficient phytoremediators of this heavy metal. Also, the Cd concentrations applied cannot be regarded as extreme, although they are typically inhibitory for many other species, and were earlier used several times by the authors as typical conditions for treatments of other plants.
Still, in search for potent plants applicable in Cd phytoremedial actions, the proposed model appears interesting, the data are worth publishing provided they are complemented with the missing results, and the work might bring many citations for both the authors and the Journal.
Indeed, the authors give good rationale for applying woody plants in phytoremediation, especially taking advantage of unique features of B. papyrifera. They also reveal very interesting Cd-adaptation mechanisms (e.g. growth stimulation under stress conditions, some metabolic alterations or pigment synthesis) which broaden our knowledge about general plant-abiotic stress reactions and might attract attention of plant physiologists.
The experimental work is based on properly applied standard methodology and is, in general, well documented, analyzed, presented and statistically evaluated. The experimental data are in large part presented properly, no major inconsistencies found.
The Introduction is informative and gives clear basis for research of the presented study. The references are relevant and properly cited throughout the text, although the reference list needs detailed check and correction.
All the Figs and tables are necessary and relevant to the content, and give direct data support for the Results. They are clearly described and appropriately referred to in the text.
The Discussion chapter has been written with the clear authors’ effort to be thorough enough; however, LARGE PARTS OF THIS SECTION ARE TOTALLY NON-UNDERSTANDABLE AND REQUIRE REWRITING. Similarly, Conclusions will be acceptable after language improvement.
Regrettably, the general faults are as follows:
- LACK of ASSESSMENT of the LEVEL of Cd ACCUMULATED by the plant seedlings (roots/shoots) as well as the Cd concentration in soil upon spiking with variant doses of Cd salts. Cd determination in the soil and plant material does not seem to be any problem – and such data would bring very important information on the mechanism of the plant’s tolerance/ response to the heavy metal stress.
This is particularly important in the context of such statements as the one found in Discussion, line 287 (“In this study, B. papyrifera may have a repellent effect on Cd.”) which is just a supposition if not a speculation, without any scientific proof, and could be easily verified by Cd assessment in the plant tissues, and unfortunately it was not done “in this study”.
- Also, there are numerous problems with the English language and especially with clarity of expressing main ideas. Sometimes it is difficult to understand what the authors really wanted to say and I am sure they tried to deliver some important message – therefore, these parts must be rewritten to enable the reader to understand (please look for a number of examples – e.g. in Discussion!). In many cases inappropriate words are used, wrong style/grammar/word order, singular/plural, sequence of tenses (mixing past with present, e.g. in Discussion) etc., and THUS IT IS STRONGLY ADVISABLE THAT THE AUTHORS SUBJECT THE MANUSCRIPT TO LANGUAGE PROFESSIONAL EDITING OR GET IT CHECKED BY A NATIVE SPEAKER experienced in biological sciences.
I have done some effort to help with the most straightforward cases and the respective corrections have been proposed below. Still, the number of inappropriately used words and text fragments difficult to correct is large and often goes beyond my competencies, as I am not a native speaker, either.
THE OTHER PROBLEMS REQUIRING CORRECTIONS ARE GIVEN HERE:
- The issue of “mine restoration” and the applicability of the model tree (Abstract and Conclusions) – this idea is illegitimate and poorly supported.
What data make the authors convinced about the suitability of B. papyrifera for “mine restoration” They give no clues. The evidenced tolerance is obviously not enough to make any plant as a perfect candidate for reclamation of mining areas! This is a ghost-like idea and seems to be given without any deep reasoning. There is not a word given in the Discussion section about such an opportunity. So, the statement that “Our study proved… is a good tree species for mine restoration“ is not true and should be removed.
- Plant species names have to be put in italics (e.g. Pteris vittata, Phytolacca acinosa… Caragana … Artemisia sacrorum)
- Please give clear explanation of the “CK” abbreviation, at least once, clearly; now it is unclear – as stated in l. 327: contaminated soil with Cd concentrations of 0 μmol/L (CK) [??]
- Provide the lacking lighting conditions applied for plants growth under greenhouse culture, or was it uncontrolled?
- The following interpretation is not consistent if not logical:
line 116-119:
In addition, the moisture content in root, stem and leaf of the seedlings were not affected significantly (P>0.05), the roots was 75% ~ 80%, and the stems content was 60% ~ 70%. Higher moisture content provides a support for the survival of B. papyrifera seedlings under Cd stress.
So, if it is not affected significantly, why do the authors claim it is “higher” and they try to give it as an explanation for the “survival under Cd stress”?
- line 163
Must rewrite the sentence which is totally non-understandable and too complex:
With the extension of time, the soluble sugar content increased first and then decreased with the increase of Cd concentration.
- Figure 3.
Add (A – C) marks to the panels – they have not been provided in the manuscript.
What did you mean by the expression “(The same below)” Please replace this shortcut??
- Abbreviations – their list is lacking, please provide one or give clear explanations of many abbreviations used: e.g. SOD, POD, CAT, MDA, CK, ABA…?
- Please correct the erroneous “different concentrations of Cd stress” (Figure captions and throughout the text, numerous cases)
The is no such a quantity as a “CONCENTRATION OF STRESS”
- References
The formal manner of citing is inconsistent and cannot be accepted as it is now. The authors must check all items in detail, and keep to one assumed convention. Now it is often a mess. Watch for capital/low case letters in journal names, wrong abbreviations and titles (e.g. Environmental Science and Pollution Research – not Environmental science and pollution research international, as well as many such cases!)
e.g. ref. 64; Does Alia have a name? why is it cited et al – there are just 2 names! Etc., etc. The authors seem to have worked in a hurry and did not pay enough attention to correct the draft text.
OTHER language problems or misleading phrases, TECHNICAL ISSUES, PROBLEMS AND MISTAKES TO BE CORRECTED
line 16
Cadmium
-> cadmium
we used pot experiment to carry out
-> we carried out a pot experiment to assess plant height…
line 19
what does CK mean – do not abbreviate in abstract
line 22
Please rewrite – the sentence is unclear
Cd had a significant inhibitory effect on chlorophyll synthesis at 12 h, while the high concentration was not significantly different from CK.
(It had an inhibitory effect but at the same time - high concentration it had not?)
line 24
which reflecting the special effect
-> which reflected the specific [??] effect
line 35
are industrial activities other than “human” ones?
line 37
and difficult to decompose
and is difficult to decompose
line 41
Hyperaccumulator
Hyperaccumulators
line 44
with super enrichment ability to zinc (Zn)
rewrite – “super ability to zinc” ? appears to make little sense
line 144
leaves
add: “leaves, respectively”.
line 223
and thus resistant to heavy metal stress
and thus resistance to heavy metal stress
line 226
Secondly … there is no “Firstly”? Check for style and consequence
line 228
what do the authors mean by “the like” in the phrase “thereby repairing stress-damaged proteins and the like” ? – must replace with something less colloquial and more meanigful
line 252
Soluble sugars … can be protected them from oxidative damage
Rewrite; this makes no sense
line 255
what is an “anti-oxidative stress”?? The cited authors clearly refer to the oxidative stress !
line 257
what is “The highest point of soluble sugar content “? And what does the rest of the phrase mean?
“The highest point of soluble sugar content shifted to high concentration with the prolongation of stress time” ?
line 283
H2O2
lines 304- 306
We believe that a slight Cd concentration could inhibit chlorophyll synthesis in B. papyrifera leaves, while a high concentration of Cd promotes the performance of other functions.
The message is unclear – what is meant by “promoting performance of other functions”? while speaking of the chlorophyll synthesis?
And not “slight” but “low” concentration, rather.
lines 307 and other
Chlorophyll a, b should be in italics
line 336
Then they were de-enzyme at 105..? de-enzyme??
Please make sure and correct
line 388
we detailed the varieties ??
must correct – the phrase is not understandable
line 390
different time Cd stress conditions
different time duration of the applied Cd stress conditions [?]
It found
It was found
Please correct:
can adapt to the damage caused by Cd stress
can adapt to the damage caused by Cd stress
line 393
in the low concentration Cd treatment although it was not as good
at low concentrations of Cd treatment although it was not as good
line 393
Is really CK within the stress group?? Or it is a control, rather? Or is it just a problem with word order – then rearrange this fragment
line 399
good tree species for mine restoration, providing a basis for future molecular biology research and development.
Please rewrite, it makes no sense – suggests that the mine restoration provides a basis for molecular biology research?? And what has been meant by “molecular biology development?”
Reviewer 3 Report
The manuscript is poor, not very interesting and appears as just another paper on Cd stress, but without any novelty or relevance to be added to the previous papers already published in this area. Additionally, the methods used are not accessible to anyone (internal reference of a school book). The statistics is also awkward and is missing in several parts either in content or details (see comments below). Moreover without Cd quantification there is no evidence that the observable changes are due to Cd entry or to other phenomenon’s like Cd/Zn co-transport or other elemental interactions.
- Why did the authors choose these concentrations? Are they relevant in potential sites to be remediated? Are they ecologically relevant?
- No Cd quantification analysis in the plant tissues.
- The authors used one-way ANOVA but there are no evidences or statements that the homogeneity and normality of the data was tested before ANOVA application.
- Anti-oxidant enzymatic activities and biomarkers were preformed according to a reference that is mostly not accessible to any reader. With hundreds of references that use well described protocols for this porpoise, this reference does not allow the readers to evaluate the analysis protocol of these enzymes. Additionally, that are very poorly described in the methods section. The same for sugar and pigment content.
- What is r/min? is it rpm?
Figure 1 – No statistical analysis
Figure 2 – The resolution of the images is very low and thus is difficult to identify the structures
Figure 4 – Lacks indication of what (average?) is plotted.
Figure 5 – lack information on what is plotted (average?) and on the statistical significance